# Increasing evidence of mechanical force as a functional regulator in smooth muscle myosin light chain kinase

Fabian Baumann[1], Magnus Sebastian Bauer[1,2], Martin Rees[3], Alexander Alexandrovich[3], Mathias Gautel[3], Diana Angela Pippig[1], Hermann Eduard Gaub[1]*

[1]Chair for Applied Physics and Center for Nanoscience, Ludwig-Maximilians-Universität München, Munich, Germany; [2]Center for Integrated Protein Science Munich, Ludwig-Maximilians-Universität München, Munich, Germany; [3]Randall Division of Cell and Molecular Biophysics, King's College London BHF Centre of Research Excellence, London, United Kingdom

*For correspondence: gaub@lmu.de

**Abstract** Mechanosensitive proteins are key players in cytoskeletal remodeling, muscle contraction, cell migration and differentiation processes. Smooth muscle myosin light chain kinase (smMLCK) is a member of a diverse group of serine/threonine kinases that feature cytoskeletal association. Its catalytic activity is triggered by a conformational change upon $Ca^{2+}$/calmodulin ($Ca^{2+}$/CaM) binding. Due to its significant homology with the force-activated titin kinase, smMLCK is suspected to be also regulatable by mechanical stress. In this study, a CaM-independent activation mechanism for smMLCK by mechanical release of the inhibitory elements is investigated via high throughput AFM single-molecule force spectroscopy. The characteristic pattern of transitions between different smMLCK states and their variations in the presence of different substrates and ligands are presented. Interaction between kinase domain and regulatory light chain (RLC) substrate is identified in the absence of CaM, indicating restored substrate-binding capability due to mechanically induced removal of the auto-inhibitory regulatory region.

## Introduction

All cells need to withstand as well as actively generate forces during division, differentiation or for their differentiated function. These mechanically governed adaptive processes require the translation of mechanical signals into physicochemical signals inside the cell that trigger an appropriate biological response (*Shivashankar et al., 2015*). A number of mechanosensors have evolved for these purposes, including mechanosensitive ion channels. The cytoskeleton, comprising microtubules, intermediate filaments and contractile actin-myosin filaments, plays a key role in maintaining cell shape against internal and external forces, but also emerges as a main hub for mechanosignaling. The regulation of actomyosin contraction differs significantly between the calcium-activated striated muscles and myosin-light chain phosphorylation in smooth and non-muscle cells. Four myosin light chain kinases (MLCK) exist, transcribed from the MYLK1 to 4 genes (reviewed in *Chang et al., 2016*).

Smooth muscle myosin light chain kinase (smMLCK) is a ubiquitously expressed serine/threonine kinase that particularly contributes to the regulation of smooth muscle contraction (*Gallagher et al., 1997*; *Hong et al., 2011*). SmMLCK phosphorylates the regulatory light chain (RLC) of smooth muscle myosin, which in turn triggers ATPase activity of the myosin heads. This activation results in the myosin power stroke – the fundamental process of muscle contraction (*Dillon et al., 1981*). Different

MLCK homologues exist for skeletal and cardiac muscle tissue, but only in smooth muscle cells the contraction is directly linked to the activity of smMLCK (*Zhang et al., 2010*). Other regulatory proteins such as troponin are not present in smooth muscle cells. In contrast to skeletal or cardiac MLCK, smMLCK is not restricted to muscle tissue but is expressed in almost all mammalian cells. Further studies have revealed the importance of smMLCK in additional cellular pathways besides muscle contraction such as platelet aggregation, exocytosis, and cell migration (*Hashimoto et al., 1994*; *Kumakura et al., 1994*; *Chen et al., 2014*).

The most important regulator of smMLCK activity is $Ca^{2+}$-loaded calmodulin ($Ca^{2+}$/CaM). Without external activation, smMLCK's catalytic turnover is suppressed by a pseudosubstrate mechanism, i.e. its catalytic core is auto-inhibited by a regulatory element that mimics the RLC substrate sequence (*Pearson et al., 1988*). Binding of $Ca^{2+}$/CaM, however, initiates a conformational change in this regulatory region and thus removes auto-inhibition. Release of the regulatory sequence facilitates RLC binding and catalytic activity of smMLCK (*Bagchi et al., 1992*).

Several isoforms are encoded by the smMLCK gene (MYLK1) by alternative initiation sites and differential splicing (*Lazar and Garcia, 1999*), containing an N-terminal extension in addition to the core catalytic kinase domain. The core sequence is highly conserved in its domains and in their respective order. It comprises immunoglobulin-like domains, a fibronectin-like domain, a proline-rich presumably elastic region as well as the catalytic kinase domain. By homology with other members of the MLCK family (*Chang et al., 2016*), the kinase domain is composed of a smaller N-terminal lobe that contains the ATP binding site and a larger C-terminal lobe that is responsible for substrate recognition (*Gallagher et al., 1997*). The latter is auto-inhibited with respect to RLC binding by the regulatory element with adjacent CaM binding region. The active site is located at the interface between the two lobes. So far, the molecular structure of smMLCK and how it conveys this regulation have not been fully elucidated. However, a high degree of structural similarity to titin kinase (TK) and twitchin kinase – two prominent serine/threonine kinases from the giant muscle protein titin and the titin-like protein twitchin found in invertebrate muscles – as well as to the recently solved MYLK4 can be assumed (*Gautel, 2011*; *Chang et al., 2016*). The terminal domains of smMLCK are also highly conserved and form binding regions to F-actin (N-terminal) and myosin (C-terminal) (*Sellers and Pato, 1984*; *Hong et al., 2009*; *Gautel, 2011*); longer isoforms (MLCK-210) also seem to interact with other cytoskeletal components through their N-terminal domains (*Kudryashov et al., 2004*). Specific binding to myosin presumably increases the affinity between the kinase domain and RLC due to local proximity (*Silver et al., 1997*). The actin-binding domain allows smMLCK to associate along actin filaments and thus enhances its phosphorylation rate in smooth muscle (*Hong et al., 2015*). The fact that these binding sites are located at the termini of the molecule suggests that smMLCK might connect simultaneously with both myosin and actin, and is theoretically capable of bridging thick and thin filaments in smooth muscle due to extensible linker regions in the proline rich repeat segment (*Mabuchi et al., 2010*). This cytoskeletal association of smMLCK could hence significantly contribute to stiffness and passive tension of smooth muscle, or to responses in external stress. An intriguing interplay exists between smMLCK and mechanical forces in some tissues; for example, repeated contractile activation leads to increased contractility in airway smooth muscle (*Fairbank et al., 2008*). Such mechanosensitive conformational modulation might be comparable to the role of TK in striated muscle cells. Titin bridges the thick and thin filament systems in the sarcomere in an analogous manner to smMLCK. It may thus act as a muscle mechanosensor, signaling exposure to mechanical tension in the contracting and mechanically stressed sarcomere. TK is also intrasterically regulated by a pseudosubstrate mechanism, but its auto-inhibition is understood to be released upon mechanical stress rather than allosterically by an effector molecule. Single-molecule studies as well as molecular dynamics simulations established that partial unfolding of TK upon external force results in a controlled release of the regulatory segment (*Puchner et al., 2008*; *Gräter et al., 2005*). This process forms an enzymatically active intermediate capable of ATP binding and subsequent substrate turnover. $Ca^{2+}$/CaM affinity has also been detected for TK, but its binding shows only a stimulating effect rather than a full activation of its turnover (*Mayans et al., 1998*). Due to these striking similarities between smMLCK and TK both in structure and in their actin-myosin association in the muscle, the existence of a comparable $Ca^{2+}$/CaM-independent regulation mechanism for smMLCK is plausible (*Chang et al., 2016*).

While the established $Ca^{2+}$/CaM activation mechanism is the most prominent and best-understood for the activation of smMLCK, other activation/regulatory factors are likely to exist, some of

which are already identified (*Stull et al., 1993*; *Pfitzer, 2001*). In this study, the mechanical response of smMLCK was probed via single-molecule force spectroscopy with an atomic force microscope (AFM) to understand how this cytoskeletal kinase is conformationally modulated by external forces. We investigated the effects of the presence of different ligands such as ATP, $Ca^{2+}$/CaM or RLC peptide substrate or a combination of these on smMLCK's pathway through its different conformational states until fully unfolded. Ligand interactions and the resulting changes to the smMLCK energy potential give further insights into its intrasteric regulatory mechanism and whether substrate binding can be enabled by the application of mechanical force.

## Results

### Force response of smMLCK during AFM-based force spectroscopy

The investigated molecular construct is a truncated version of the smooth muscle isoform encoded in the smMLCK gene (*Lazar and Garcia, 1999*) lacking the proline rich region and the N-terminal actin-binding domain. The remaining sequence comprises the kinase domain, its neighboring fibronectin-like domain (Fn3) and two N-terminal Ig-like domains ($Ig_1$, $Ig_2$) as well as the C-terminal Ig-like myosin-binding domain called telokin ($Ig_T$) (*Figure 1A*). For force spectroscopy experiments, the smMLCK is specifically tethered by an N-terminal Strep-tag II via an AFM cantilever tip that is functionalized with a monovalent variant of Strep-Tactin (*Baumann et al., 2016*). Force is applied in physiological pulling geometry with the PEG spacers minimizing unspecific protein-surface interactions.

When the molecular construct is stretched with a constant speed, the investigated protein passes through a characteristic sequence of conformational states. These protein states represent semistable folding intermediates on the guided way through the protein's energy potential starting from a fully folded native structure. The conformations might also correspond to partially unfolded structures which are stable enough to form functional states. The transitions between these states are marked by distinct drops in the force-distance curves. In all cases reported here these transitions are accompanied by a lengthening of the molecular construct, which is manifested in a characteristic release of hidden contour length. The recorded force-distance response of smMLCK reveals a strict hierarchy in mechanical stability of the individual domains (*Figure 1B*). At low forces, the kinase domain unfolds by passing an intermediate conformational state $S_2$ indicated by two force peaks at the end of state $S_1$ at ~30 nm and of $S_2$ at ~60 nm. Both transitions appear at forces of approximately 30 pN, measured at a retraction speed of 800 nm/s. At state $S_3$ the kinase domain is fully unfolded. This characteristic sequence is followed by Fn3 unfolding, which is assigned to a single rupture event at around 100 pN in accordance with the respective contour length and previous data on domain strength (*Rief et al., 1998*; *Li et al., 2005*). Finally, the three Ig-like domains in the construct denature at forces of around 200–250 pN (*Rief et al., 1998*). Significant elongation of the surrounding linker length due to preceding kinase unfolding does not alter rupture forces of these latter domains in our study and is therefore unlikely to influence the hierarchy of rupture forces in the probed structure or its characteristic sequence of unfolding. Since the employed tethering complex (Strep-tag II:Strep-Tactin) substantially overlaps in its rupture force regime with the unfolding force of Fn3, most force distance traces end either before or after the Fn3 domain because the tethering complex ruptured before the Ig-like domains could unfold. A transformation of 99 force-distance curves into a contour length histogram (*Figure 1C*, *Figure 1—source data 1*) indicates that the rarely unfolded Ig-like domains and Fn3 contribute a contour length of 30 nm each. These increments are consistent with results obtained in other force spectroscopy measurements (*Rief et al., 1998*). The kinase domain is assigned with respect to its contour length to a rather clear peak at $61.7 \pm 9.6$ nm ($L_{2 \rightarrow 3}$ released at transition $S_2 \rightarrow S_3$) and a less distinct broad ridge in the range of 25–40 nm ($L_{1 \rightarrow 2}$ released at transition $S_1 \rightarrow S_2$). The relatively wide contour length distribution of the broad ridge may be caused by the low force of this event and the consequently poor transformation of this segment: further aspects which might contribute to this poorly defined transformation are discussed in the context of the ATP-binding measurements below. For an estimation of the released contour length at $S_1 \rightarrow S_2$ transition, individual force-distance curves with high rupture forces at this point were fitted with the Worm-Like Chain (WLC) model yielding an approximate length increment of around 30 nm. The measured contour length of approximately 92 nm for the full kinase domain is

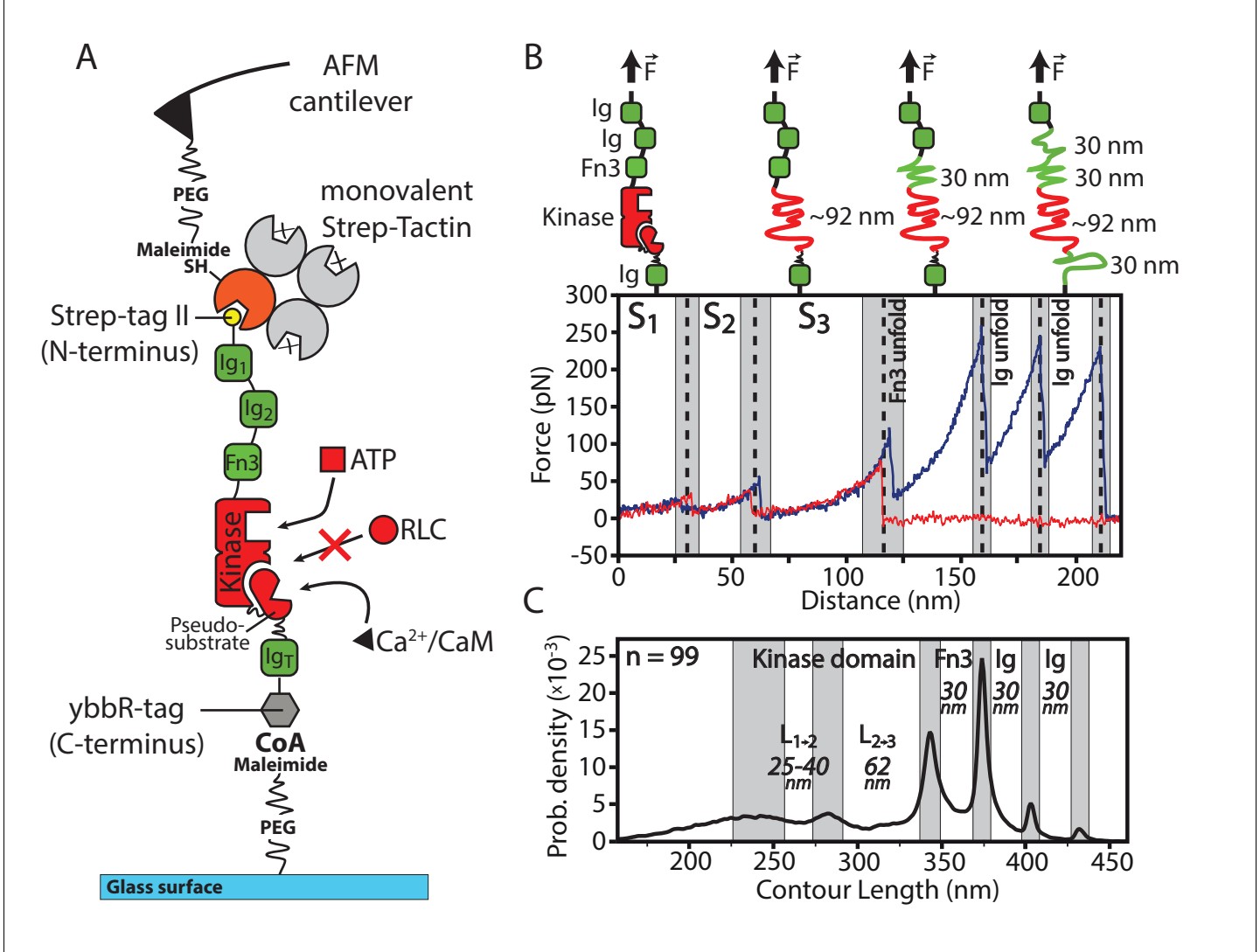

**Figure 1.** Overview of the experimental configuration for applying controlled mechanical stress to smMLCK. (**A**) Schematic illustration of the investigated smMLCK construct. It consists of the kinase domain surrounded by several Ig-like domains (Ig) and a fibronectin-like domain (Fn3). Possible substrate interactions are indicated (ATP, $Ca^{2+}$/CaM and RLC). RLC interaction is prevented by the auto-inhibitory pseudosubstrate sequence that is released upon $Ca^{2+}$/CaM binding. For covalent attachment onto the surface, the construct harbors a C-terminal ybbR-tag. (**B**) Representative force-distance curves (red, blue) depicting the characteristic transitions of the kinase through different conformational states ($S_1$, $S_2$, $S_3$) and subsequent unfolding of the adjacent Fn3 and Ig-like domains. Whereas most force-distance curves rupture before or after Fn3 unfolding (as shown in red) due to comparable rupture forces of Fn3 and the employed handle system, the blue curve illustrates a descriptive example with additional unfolding of Fn3 and Ig-like domains depicting the further force-distance pattern given by the construct. Structural interpretation and assignment of the detected force-distance pattern is schematically depicted above the curve. (**C**) Contour length transformation of 99 unfolding events with respective contour length increments. $L_{1\to2}$ and $L_{2\to3}$ are released at the transition of the kinase domain from conformational state $S_1$ to $S_2$ and $S_2$ to $S_3$ respectively. The contour lengths of Fn3 and Ig-like domains are additionally depicted.

The following source data is available for figure 1:

**Source data 1.** Contour length plot of 99 unfolding events of MLCK with 0 mM ATP present, aligned as described in the data analysis section.

in good agreement with a simple approximation assuming 0.365 nm per amino acid (255 aa ×0.365 nm = 93.1 nm) (*Dietz and Rief, 2004*). It indicates a fully unfolded kinase domain at $S_3$. The two-step unfolding behavior is consistent with the bi-lobed structure of the kinase's catalytic core and

suggests assignment of $S_1 \rightarrow S_2$ transition to the unfolding of the smaller kinase lobe and of $S_2 \rightarrow S_3$ to the larger lobe.

## Conformational changes of smMLCK upon ATP binding

In contrast to TK, smMLCK's capability of binding ATP is not inhibited through its pseudosubstrate mechanism. Independent from $Ca^{2+}$/CaM activation, ATP interacts with a $K_d$ of around 10 µM (*Kennelly et al., 1992*). In this study, ATP binding to smMLCK and the corresponding effects on its structure were identified by changes in the characteristic transition pattern through the kinase's different conformational states during AFM-based force spectroscopy. ATP was added in buffer solution with a final concentration of 3 mM. A heatmap in force-distance space of 560 aligned and overlaid unfolding curves emphasizes ATP-induced changes: the smaller kinase domain lobe ($S_1 \rightarrow S_2$ transition) is significantly stabilized upon ATP binding (*Figure 2A*). Since the order of released contour lengths $L_{1 \rightarrow 2}$ and $L_{2 \rightarrow 3}$ – associated with the transitions $S_1 \rightarrow S_2$ and $S_2 \rightarrow S_3$ – is not altered by this enhanced stability, the $S_2 \rightarrow S_3$ transition seems to remain structurally shielded from $S_1 \rightarrow S_2$. The increased force signal of $S_1 \rightarrow S_2$ allows precise extraction of the small lobe increment $L_{1 \rightarrow 2}$ in contour-length space. The determined $30.8 \pm 9.8$ nm add to a total length of $92.6 \pm 7.5$ nm for the full kinase domain with $L_{2 \rightarrow 3}$ being unchanged with a contour length of $61.8 \pm 8.9$ nm (*Figure 2B*). Other contour length increments in the overall construct including Fn3 and Ig-like domains were unaffected by the interaction of kinase domain and ATP. Unfolding traces in absence and presence of ATP were realized within one experiment that is, same cantilever and sample surface. The measured forces are therefore directly comparable without uncertainties in the force signal that could originate from deviations in AFM spring constant calibration. The histograms for the unfolding force of the $S_1 \rightarrow S_2$ transition with and without ATP indicate an increase of the most probable rupture force of about 30 pN (*Figure 2C*). Due to saturated binding conditions using 3 mM ATP, the obtained force histogram most likely represents exclusively the ATP-bound conformation and not a mixed population of ATP-bound and ATP-free states of the tethered smMLCK molecules. The peak forces for $S_2 \rightarrow S_3$ and Fn3 were found to be unaffected by addition of ATP (*Figure 2—figure supplement 1*). The fact that Fn3 remains unchanged in its unfolding properties independent of ligand addition allows for relative comparison of measurements from different cantilevers. In the following, forces from different experiments are normalized according to the most probable peak force of Fn3 unfolding. Binding of ADP as well as the non-hydrolysable ATP analogue adenylyl-imidodiphosphate (AMP-PNP) could analogously be detected for smMLCK by a stabilization of the $S_1 \rightarrow S_2$ transition in the tethered construct (*Figure 2—figure supplement 2*). Stabilization, however, appears to be weaker in comparison to the interaction with ATP.

## Energy barrier attenuation upon $Ca^{2+}$/CaM interaction

Importantly, an additional peculiarity of the kinase unfolding becomes evident from the ATP-stabilized case: an unusual stretching behavior of the $S_1$ state at AFM distances of about 30–40 nm (*Figure 2A*). The force response in this region clearly deviates from a typical WLC behavior exhibiting an unusual kink. Apparently, a small but distinct energy barrier - a mechanical barrier stabilizing a specific protein conformation - has to be overcome at low forces (~15 pN) before the kinase domain acquires conformational state $S_1$. The energy barrier is attributed to an additional transition $S_0 \rightarrow S_1$ preceding the formerly described transitions $S_1 \rightarrow S_2$ and $S_2 \rightarrow S_3$ (*Figure 3A*). $S_0 \rightarrow S_1$ represents a small, uncoupled conformational change with a barely resolvable contour length release. This additional energy barrier might also explain the ambiguous contour length transformations for the $S_1 \rightarrow S_2$ transition in the measurements in the absence of ATP (*Figure 1C*). Due to the small forces of transitions $S_0 \rightarrow S_1$ and $S_1 \rightarrow S_2$ they are hard to distinguish in the unfolding pattern and their transformations result in a broadened overlap of their respective distributions.

The existence of this energy barrier becomes particularly relevant as the presence of $Ca^{2+}$/CaM affects this initial feature. Namely, addition of $Ca^{2+}$/CaM to the measurement buffer leads to an absence of this barrier otherwise observed in the kinase force response (*Figure 3A*). $Ca^{2+}$/CaM is understood to be a trigger for structural rearrangements in the kinase domain that enable RLC binding. The detected energy barrier could represent this rearrangement mechanism. We thus propose the following hypothesis: the release of the pseudosubstrate is already realized upon initial $Ca^{2+}$/CaM binding to smMLCK and is therefore not observed. If not activated by $Ca^{2+}$/CaM, however, this

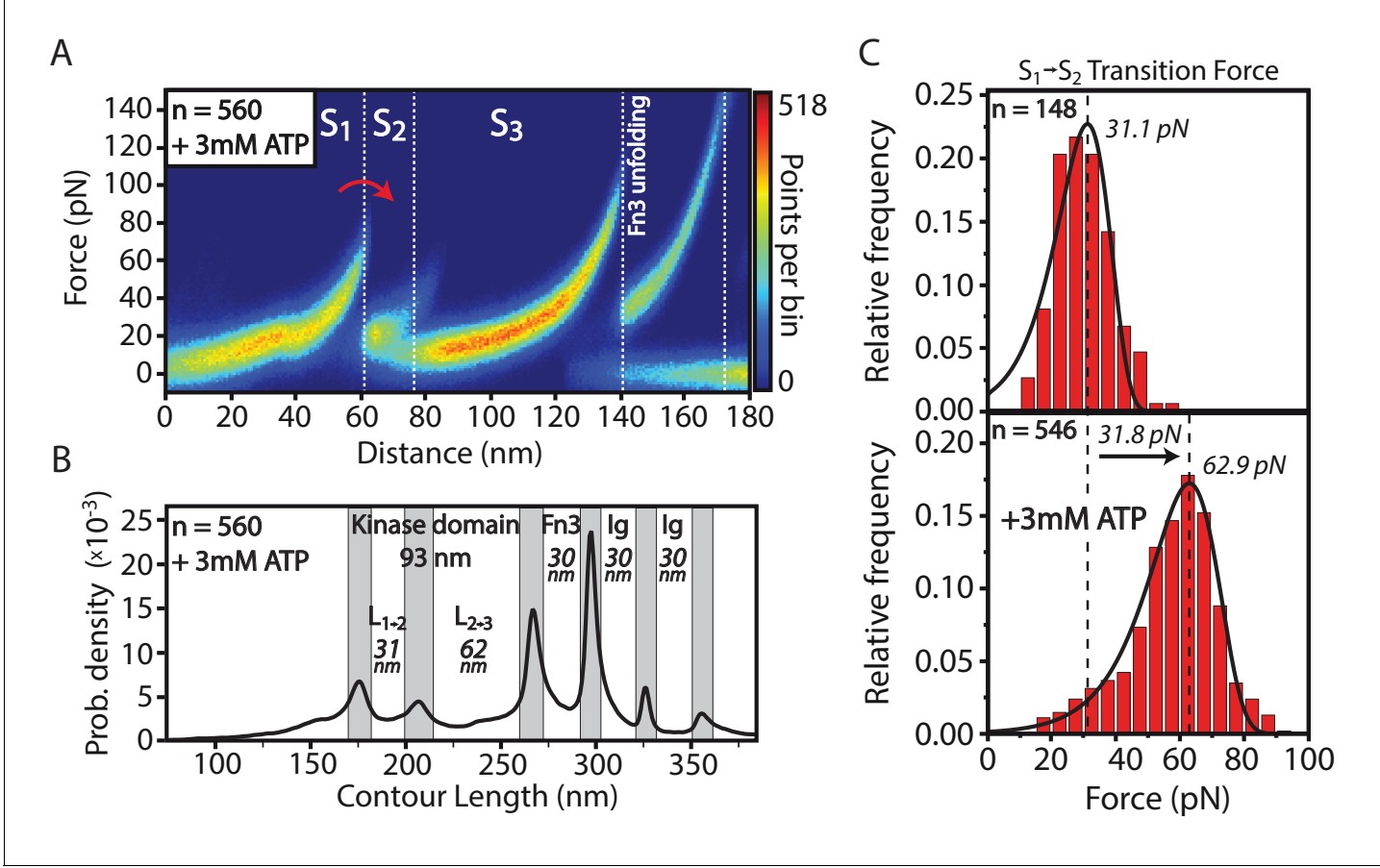

**Figure 2.** Structural effects of ATP binding on smMLCK's characteristic sequence of conformational states. (**A**) Stabilization of the $S_1 \rightarrow S_2$ transition upon ligand binding. For better illustration, a heatmap of 560 aligned curves is depicted. (**B**) Contour-length transformation of the respective events in the presence of ATP. $L_{i \rightarrow j}$ is associated to the contour length released at the transition from state $S_i$ to $S_j$. (**C**) Statistical evaluation of $S_1 \rightarrow S_2$ stabilization via force histograms fitted with the Bell-Evans model. An increase in the most probable transition force of about 30 pN is observed upon ATP addition. Both data sets were recorded within one experiment with the same cantilever.

The following source data and figure supplements are available for figure 2:

**Source data 1.** Contour length plot of 560 unfolding events of MLCK in the presence of 3 mM ATP, aligned as described in the data analysis section.

**Source data 2.** Force histogram of $S_1 \rightarrow S_2$ transition in the presence of 0 mM ATP.

**Source data 3.** Force histogram of $S_1 \rightarrow S_2$ transition in the presence of 3 mM ATP.

**Figure supplement 1.** Effects of ATP or $Ca^{2+}$/CaM addition on the peak forces for the respective transitions $S_1 \rightarrow S_2$ and $S_2 \rightarrow S_3$ and for the Fn3 unfolding force.

**Figure supplement 1—source data 1.** Force histogram of $S_1 \rightarrow S_2$ transition in the presence of 0 mM ATP.

**Figure supplement 1—source data 2.** Force histogram of $S_1 \rightarrow S_2$ transition in the presence of 3 mM ATP.

**Figure supplement 1—source data 3.** Force histogram of $S_1 \rightarrow S_2$ transition in the presence of 3 mM ATP, 25 μM CaM, 2 mM $Ca^{2+}$.

**Figure supplement 1—source data 4.** Force histogram of $S_2 \rightarrow S_3$ transition in the presence of 0 mM ATP.

**Figure supplement 1—source data 5.** Force histogram of $S_2 \rightarrow S_3$ transition in the presence of 3 mM ATP.

**Figure supplement 1—source data 6.** Force histogram of $S_2 \rightarrow S_3$ transition in the presence of 3 mM ATP, 25 μM CaM, 2 mM $Ca^{2+}$.
*Figure 2 continued on next page*

*Figure 2 continued*

**Figure supplement 1—source data 7.** Force histogram of Fn3 unfolding in the presence of 0 mM ATP.

**Figure supplement 1—source data 8.** Force histogram of Fn3 unfolding in the presence of 3 mM ATP.

**Figure supplement 1—source data 9.** Force histogram of Fn3 unfolding in the presence of 3 mM ATP, 25 µM CaM, 2 mM $Ca^{2+}$.

**Figure supplement 2.** Stabilization of the $S_1 \rightarrow S_2$ transition upon ADP or AMP-PNP binding.

**Figure supplement 2—source data 1.** Force histogram of $S_1 \rightarrow S_2$ transition in the presence of 3 mM AMP-PNP, 30 µM CaM, 3 mM $Ca^{2+}$, 280 µM RLC.

**Figure supplement 2—source data 2.** Force histogram of Fn3 unfolding in the presence of 3 mM AMP-PNP, 30 µM CaM, 3 mM $Ca^{2+}$, 280 µM RLC is used for normalizing forces to the same value.

**Figure supplement 2—source data 3.** Force histogram data of $S_1 \rightarrow S_2$ transition in the presence of 4 mM ADP.

**Figure supplement 2—source data 4.** Force histogram of the Fn3 unfolding in the presence of 4 mM ADP is used for normalizing forces to the same value.

regulatory modulation of the smMLCK structure appears as distinct part of the transition pathway indicated by the $S_0 \rightarrow S_1$ transition (*Figure 3B*). Addition of $Ca^{2+}$ without CaM has no effect on the observed feature (*Figure 3—figure supplement 1*). Since the energy barrier precedes the complete process of guiding smMLCK through different conformations, the regulatory fragment appears to be released before the intact kinase domain gets denatured at all and therefore becomes inactive. This mechanism could therefore imply the existence of intermediate smMLCK conformations ($S_1$, $S_2$) with an intact active site and RLC binding capability independent of $Ca^{2+}$/CaM induced by mechanical forces comparable to the regulation of TK.

## RLC peptide interaction

Based on these considerations, we designed experiments to test if conformational changes in the unfolding pattern can also be detected in the presence of RLC substrate. A minimal MLCK substrate peptide was used as RLC substitute that interacts only with the catalytic core of the kinase domain (*Knighton et al., 1991*). It consists of a truncated and slightly modified smooth muscle myosin light chain (MLC 11–23, P14A, Q15A) from chicken gizzard with the amino acid sequence KKRAAR-ATSDVFA. It is effectively phosphorylated by smMLCK with a turnover rate of $K_M$ = 7.5 µM (for chicken gizzard smMLCK) (*Kemp and Pearson, 1985*). Assuming an effectively blocked RLC binding site without activation by $Ca^{2+}$/CaM, RLC addition should in principle not influence the structure of the smMLCK kinase domain due to the lack of interaction – unless there is binding by an active intermediate during the process of unfolding.

The characteristic patterns of transitions through different smMLCK states in the presence of RLC peptide are plotted via heatmaps of smMLCK force-distance curves in presence (n = 1013) and absence (n = 864) of ATP in the buffer solution in *Figure 4A*. The heatmaps clearly reveal a significant stabilization of the $S_2$ state independent of ATP. This structural change clearly indicates interaction with the substrate and is thus interpreted as RLC binding. Quantitative evaluation of $S_2 \rightarrow S_3$ transition force histograms was performed via kernel-density functions, since the recorded forces show deviations from a single-bond Bell-Evans distribution (*Figure 4B*). This might partly be based on the overlap of different populations e.g. RLC-bound and RLC-free kinase domain, but deviations from the typical Bell-Evans model are also observed for $S_2 \rightarrow S_3$ in measurements without RLC substrate. Other factors such as interactions between the catalytic domain and adjacent Fn3-like domain or the pseudosubstrate sequence could be examples that already promote different transition pathways and thus yield this atypical behavior. Independent of shape, however, the recorded histograms reveal a clear shift in most probable rupture force in $S_2 \rightarrow S_3$ transition by about 30 pN.

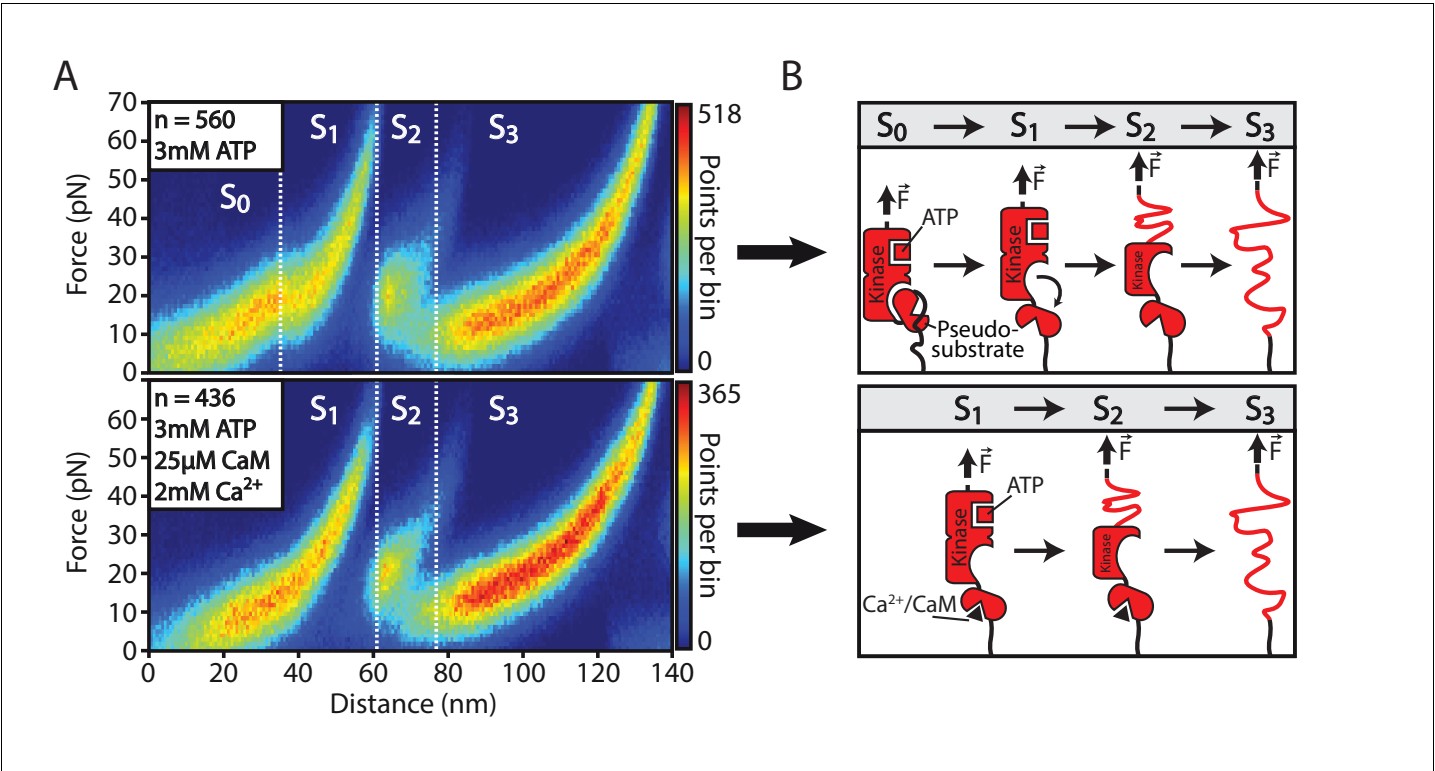

**Figure 3.** Structural effects of Ca$^{2+}$/CaM binding on smMLCK's characteristic sequence of conformational states. (**A**) Attenuated S$_0$→S$_1$ transition in the characteristic force-distance pattern of the smMLCK construct due to conformational changes upon Ca$^{2+}$/CaM binding. This effect is emphasized by a heatmap comparison of several hundred overlaid force-distance curves. Both data sets were collected within one measurement. (**B**) Structural model interpretation. The S$_0$→S$_1$ transition is assigned to a force-induced rearrangement in the kinase domain that correlates with the conformational changes induced by Ca$^{2+}$/CaM binding – the release of the inhibitory pseudosubstrate.

The following figure supplement is available for figure 3:

**Figure supplement 1.** Missing effects by addition of Ca$^{2+}$ without CaM.

## Discussion

Binding of RLC peptide is directly observed in our measurements in the absence of Ca$^{2+}$/CaM as it clearly alters the characteristic transition pattern of single smMLCK molecules by stabilizing the S$_2$→S$_3$ transition. This observation is in conflict with the blocking of the RLC binding site without Ca$^{2+}$/CaM activation as proposed by the established pseudosubstrate inhibition mechanism (*Bagchi et al., 1992*). The experimental results could therefore represent a first indicator for a force-driven activation of the catalytic pathway of smMLCK (*Figure 5*). In the living organism, we assume that smMLCK bridges actin and myosin filaments in smooth muscle tissue and thus allows for a stretched conformation that additionally promotes kinase activity. By relative movements in the cytoskeleton, sufficient mechanical stress may be created within the protein regulating its catalytic activity besides Ca$^{2+}$/CaM binding, as proposed in this study. Even for mechanical forces below 15 pN - the force that is identified to be necessary for activation in our experiments - steady physiological stress could likely remove autoinhibition if (semi-)permanently applied as the kinase is spanned between two filaments instead of being probed in a constant speed single-molecule force spectroscopy experiment. Despite a K$_d$ in the μM range between smMLCK and myosin or actin (*Hong et al., 2015*), long-term association to the cytoskeleton is assumed for smMLCK especially due to several additional binding domains to both filaments in its long isoform (*Kudryashov et al., 2004*). Necessary forces and lifetimes of the spanned conformation could therefore in principle be reached in a physiological context to allow for the proposed force-driven activation. The *in vivo* existence and relevance of this regulation pathway, however, has to be examined in further studies. Our conclusions

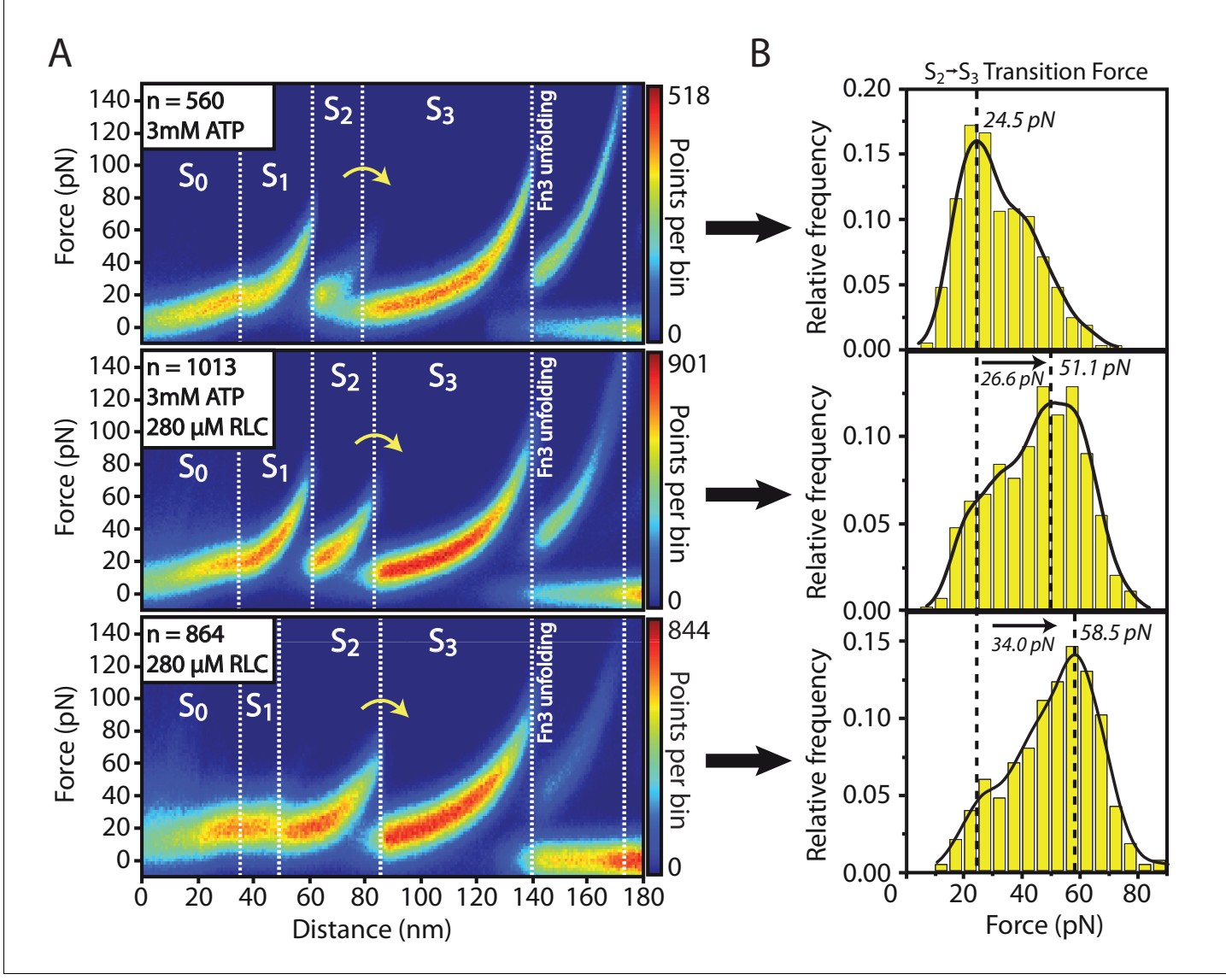

**Figure 4.** Structural effects of RLC peptide binding on smMLCK's characteristic sequence of conformational states. (**A**) Qualitative observation of an increased mechanical stability in the large kinase lobe illustrated by the higher forces in the $S_2 \rightarrow S_3$ transition. The effect is emphasized by heatmaps of aligned force-distance curves obtained under different substrate conditions. The stabilizing effect is detected independently of the presence of ATP. (**B**) Quantitative evaluation of the increased $S_2 \rightarrow S_3$ transition force. The force histograms were approximated with a kernel-density function for extracting the most probable rupture force. It reveals a significant shift of about 30 pN due to the stable interaction of the RLC peptide with the catalytic core. Since this binding is stated to be prevented by an auto-inhibition process according to the conventional view of smMLCK activation, the experimental observation hints at an additional path of kinase regulation modulated by force.

The following source data and figure supplement are available for figure 4:

**Source data 1.** Force histogram of $S_2 \rightarrow S_3$ transition in the presence of 3 mM ATP.
**Source data 2.** Force histogram of $S_2 \rightarrow S_3$ transition in the presence of 3 mM ATP and 280 μM RLC.
**Source data 3.** Force histogram of $S_2 \rightarrow S_3$ transition in the presence of 280 μM RLC.
**Figure supplement 1.** $Ca^{2+}$/CaM-dependent RLC phosphorylation of the investigated smMLCK construct.

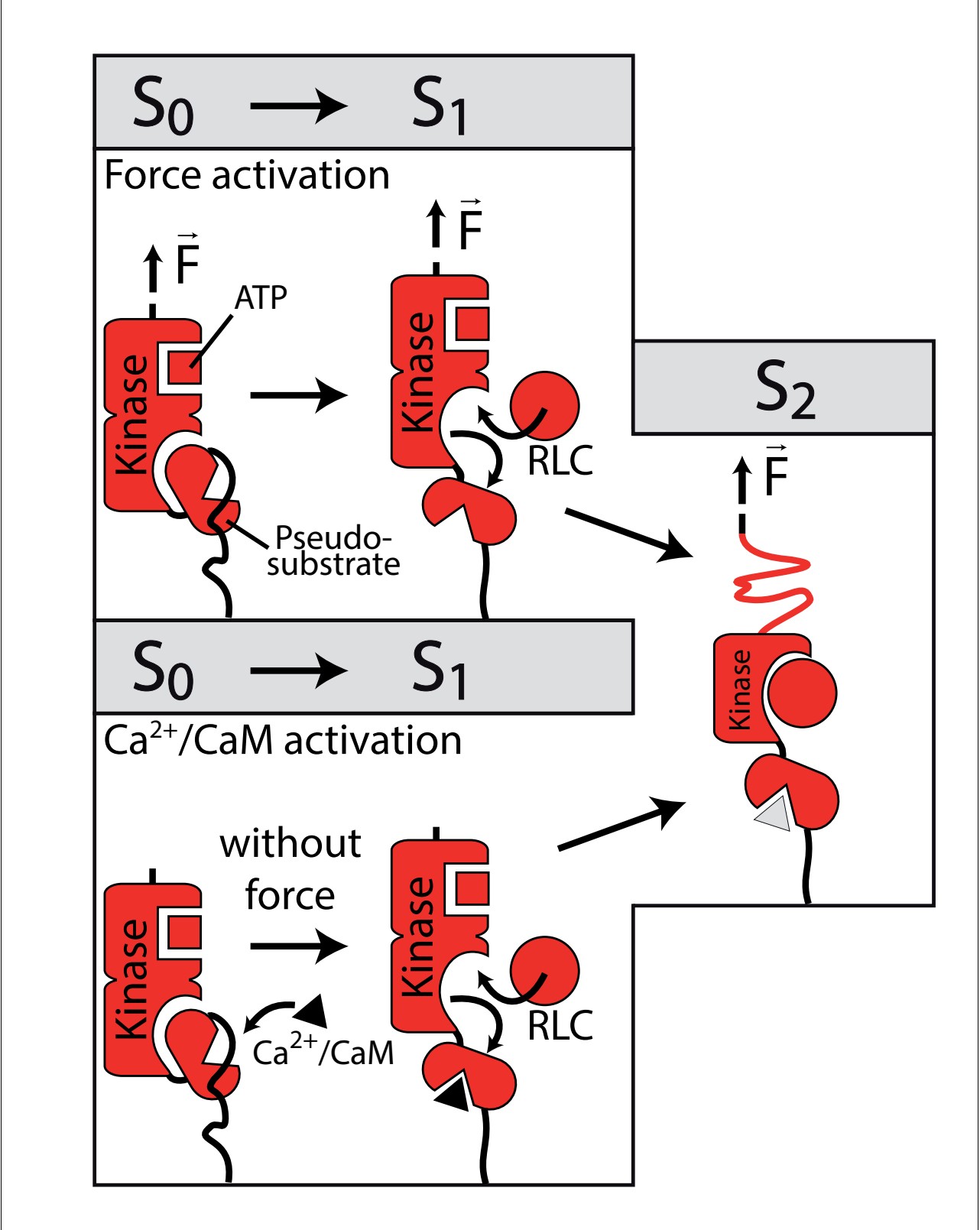

**Figure 5.** Structural interpretation of the stabilized $S_2$ state upon RLC interaction. Mechical stress forces the construct into a conformational state $S_1$ equivalent to the state reached by $Ca^{2+}$/CaM binding. By release of the pseudosubstrate sequence the conformational state is capable of RLC binding which is detected by a significant stabilization of the $S_2$ state. Both initially different activation pathways eventually result in the same sequence of conformational states, with the only difference being the presence or absence of bound $Ca^{2+}$/CaM, depicted in light grey in the $S_2$ state.

are drawn from single-molecule force spectroscopy measurements where we specifically tethered smMLCK and forced it through several conformations. The presented AFM-based approach represents a sensitive means of detecting ligand binding on the single-molecule level but can provide only limited temporal information about the interaction. From the barrier pattern, it is not directly discernible if the substrate binds in the course of the pulling (as assumed for the RLC) or if it is bound right from the beginning of measurements (as for ATP binding). In order to exclude pre-binding of the substrate, basal binding activity of RLC in absence of $Ca^{2+}$/CaM activation was tested for the used construct via isothermal titration calorimetry. Due to the presumably very low binding affinities, interaction kinetics could not be detected at moderate concentrations to distinguish between basal and mechanically induced binding. The investigated construct does not show $Ca^{2+}$/CaM-independent enzymatic activity, suggesting that RLC does not bind to smMLCK in the absence of $Ca^{2+}$/CaM under zero tension (*Figure 4—figure supplement 1*). While beyond the scope of this study, hybrid approaches will be required to obtain a complete and conclusive picture of the binding mechanism and its physiological relevance, ideally complemented by structural information of the auto-inhibited state. In particular, combined force and fluorescence spectroscopy and molecular dynamics simulations will aid in this. Ultimately, we need to develop an assay that directly visualizes substrate turnover upon force-activation of these enzymes, as *in-situ* unbinding forces from actin and myosin filaments (which are yet unknown) will interplay with the mechanically induced intramolecular conformational changes. Direct measurements of force-induced substrate binding and activity, that is, RLC phosphorylation, will then lead to full comprehension of this alternative activation path for smMLCK. To this end, the powerful combination of single-molecule force spectroscopy and fluorescence spectroscopy in nanoapertures can provide the basis for *in vitro* force-activation assays (*Heucke et al., 2013*), to complement the presented findings that RLC substrate binds smMLCK under force in the absence of $Ca^{2+}$/CaM.

## Materials and methods

### Expressed construct

The smMLCK construct used for this study is an 858 amino acid protein (808 aa, from 1097 to 1904 in human smooth muscle isoform 1 myosin light chain kinase, accession number NP_444253.3) incorporating Strep-Tag II (WSHPQFEK) at the N-terminus and a ybbR-tag (DSLEFIASKLA) and hexa-histidine tag at the C-terminus. The cDNA encoding smMLCK was cloned into the EcoRI and XhoI sites of a modified pENTR11 (Invitrogen) baculovirus shuttle plasmid region surrounded by attL1 and attL2 recombination sites (5'- attL1 - TCG AAG GAG ATA GAA CCA ATT CTC TAA GGA AAT ACT TAA CCA TGG CTA GCT GGA GCC ACC CGC AGT TCG AAA AAG GCG CCG AGA CCG CGG TCC CGA ATT CG - smMLCK - CCC TCG AGC GGT TCC GGT GGT GAC TCC CTG GAG TTC ATC GCT TCC AAG CTG CTT CA GGC CTG AGA GGA TCG CAT CAC CAT CAC CAT CAC TAA GAT CCG TCG AGA TAT CTA G - attL2 - 3'). After generation of recombinant virus using a BaculoDirect-Baculovirus Expression System (Invitrogen), production of smMLCK was carried out in suspension cultures of Spodoptera frugiperda Sf9 (Sf9 cells in Sf-900II SFM, Invitrogen). Cells were routinely maintained at 28°C, 100 rpm in the concentration range from $1 \times 10^6$ cells/ml to $8 \times 10^6$ cells/ml. 1 l culture of Sf9 cells at a concentration of $2.5 \times 10^6$ cells/ml was infected with smMLCK recombinant baculovirus of third generation (P3) and left growing for 3 days. Infected cells were then pelleted at $1000 \times g$ for 10 min. They then were resuspended in an ice-cold buffer containing 20 mM Tris (pH 8.0), 100 mM NaCl, 40 mM Imidazole, 14 mM 2-mercaptoethanol (buffer A) supplemented with cOmpleteEDTA-free Protease Inhibitor Cocktail (Roche) as per manufacturer's recommendations. Cells were lysed by passing the mixture three times through a $0.8 \times 40$ mm syringe needle and treated with DNase I at a final concentration of 25 μg/ml for 10 min at 4°C. After centrifugation at $4000 \times g$, supernatant was loaded onto a 2 ml His-Trap crude column (GE Healthcare Life Sciences) pre-equilibrated with buffer A. The column was then washed with 20 column volumes of buffer A and smMLCK protein was eluted with a step of buffer A containing 250 mM Imidazole. Peak fraction was purified further on HiLoad 26/600 Superdex 200 (GE Healthcare Life Sciences) equilibrated with 20 mM Tris (pH 8.0), 100 mM NaCl, 1 mM DTT. Peak fractions were pooled together and analysed on SDS PAGE for purity.

## Enzymatic activity assay

To assess CaM-dependent and independent enzymatic activity of smMLCK, the construct used in AFM experiments was incubated with human RLC (NP_291024.1) in the presence and absence of $Ca^{2+}$/CaM and the phosphorylation of RLC serine 19 was probed.

10 nM smMLCK was mixed with 10 µM RLC, 500 µM ATP in 20 mM HEPES (pH 7.5), 50 mM NaCl, 10 mM $MgCl_2$, 1 mM DTT, 500 µM ATP in the presence of either 40 nM CaM/1 mM $CaCl_2$ or 1 mM EGTA and incubated at 20°C. Samples were taken at various intervals with the reaction quenched by addition of SDS-PAGE loading buffer.

Samples containing 300 ng RLC were run on SDS-PAGE alongside BioRad Precision Plus protein marker, transferred onto nitrocellulose membrane, incubated in 5% milk at room temperature for 45 min and probed with Cell Signaling Technology antibody 3671 against phospo-myosin light chain 2 (Ser19) in 5% milk for 2 hr at room temperature. The membrane was washed with low-salt buffer (10 mM Tris (pH 7.4), 150 mM NaCl, 0.1% Tween-20) three times and then incubated with horseradish peroxidase conjugated anti-mouse IgG (Dako P0260) in 5% milk for 45 min. After washing three times in low-salt buffer the membrane was stained using the enhanced chemiluminescence method.

## Sample preparation

Glass coverslip and AFM cantilever were identically passivated for unspecific interactions using heterobifunctional succinimide-PEG-maleimide spacers (Rapp Polymere, Tübingen, Germany) with a molecular weight of 5000 Da (*Celik and Moy, 2012*). The succinimide group is attached via (3-aminopropyl)-dimethyl-ethoxysilane (APDMES, Karlsruhe, Germany) to the respective surface. The reactive maleimide group covalently conjugates to accessible thiol groups on applied bio-molecules used for specific immobilization. On the coverslip, this reaction is employed for covalent attachment of coenzyme A. The smMLCK construct harbors a C-terminal ybbR-tag (*Wong et al., 2008*) that site-selectively reacts to coenzyme A catalysed by the Sfp-synthase system (*Yin et al., 2005*, *2006*). The investigated construct additionally contains an N-terminal Strep-tag II that is pulled via a monovalent variant of Strep-Tactin in the force spectroscopy experiments (*Baumann et al., 2016*). The monovalent Strep-Tactin is engineered to contain a unique cysteine residue on its single functional subunit, which is utilized for specific immobilization onto maleimide-PEG functionalized cantilevers (Biolever Mini, Olympus, Tokyo, Japan) (*Zimmermann et al., 2010*). For the AFM experiments, 40 mM HEPES (pH 7.2) with 2 mM $MgCl_2$ and 1 mM DTT was used as measurement buffer.

## Force spectroscopy experiments

AFM force spectroscopy data was acquired on a custom-built AFM operated in closed loop mode by a MFP3D controller (Asylum Research, Santa Barbara, CA, USA). Software for the automated control of AFM head and xy-piezos was programmed in IgorPro6 (Wavemetrics, OR, USA). Strep-Tactin coated Biolever Mini cantilevers were briefly brought in contact with the sample surface and then retracted at 800 nm/s. After each recorded force-distance curve, the surface was horizontally moved in steps of 100 nm distance for iteratively probing a new position. The cantilever spring constant was calibrated using the equipartition theorem method (*Hutter and Bechhoefer, 1993*). Typically, datasets contain 30000 force-distance curves and the addition of the substrate was performed in the course of the experiment. If data was not collected within one experiment, but was directly compared via rupture force histograms, the recorded force values were normalized according to the most probable rupture force of the Fn3 domain. For this normalization, only those force-distance curves in a dataset were regarded that detached after unfolding of the Fn3 domain.

## Data analysis

Force heatmaps were assembled from all curves showing the characteristic unfolding pattern of the stressed smMLCK construct. Respective force spectroscopy data was aligned in force-distance space and transformed to a heatmap based on raw data points with 750 bins per axis. Data evaluation was carried out in Python 2.7. The rupture forces were evaluated from the AFM force-distance curves utilizing a quantum mechanically corrected WLC model (*Hugel et al., 2005*). The AFM distance was corrected for cantilever bending. 20 nm force baseline after the last rupture event – typically representing detachment – was used for determining zero force in the transformation of deflection signal to force values. Force-extension data was transformed into contour length space via an inverse

worm-like chain model assuming a persistence length of 0.4 nm and a thermal energy of 4.1 pN nm (*Jobst et al., 2013*). On the transformed data a Gaussian kernel density estimate is applied with a bandwidth of 1 nm. Data set alignments in contour length space are created by the following process: the full set of transformed force-distance curves is aligned to a random curve from this data set according to least residual in cross correlation. This process results in a first superimposition which is used as a template in a second iteration of this process. Again, all contour-length transformed curves are aligned to a template curve but this time to the one formed by the first iteration. This two-step approach diminishes biasing effects given by the choice of the random curve used for initial alignment. Contour lengths of the individual domains are determined by a Gaussian fit of each determined peak and subtraction of the respective fitted means. The error of an increment is given by the standard deviations of both peaks defining the individual increment. Force-distance curves were denoised with Total Variation Denoising in order to detect rupture events as significant drops in force. For the force histograms, detected peaks in the typical region of respective domain unfolding were assigned to $S_1 \rightarrow S_2$, $S_2 \rightarrow S_3$ or Fn3. All assignments were manually double-checked - especially to reassure small rupture events close to the noise level or to delete erroneously assigned peaks. In some curves no distinct rupture event could be detected for a specific domain due to too small forces (below noise level of about 10–15 pN) and was not included to the evaluation. Rupture forces of respective domains in the unfolding pattern were binned to histograms and fitted with the Bell-Evans model yielding the most probable rupture force (*Bell, 1978*; *Evans and Ritchie, 1997*). In the case of the $S_2 \rightarrow S_3$ rupture event, kernel-density estimates with a bandwidth of 1 pN were applied to the data and used for extracting the most probable rupture force.

## Amino acid sequence: SII-smMLCK-ybbR-His

MASWSHPQFEKGAETAVPNSAPAFKQKLQDVHVAEGKKLLLQCQVSSDPPATIIWTLNGKTLKTTKFII
LSQEGSLCSVSIEKALPEDRGLYKCVAKNDAGQAECSCQVTVDDAPASENTKAPEMKSRRPKSSLPPVLG
TESDATVKKKPAPKTPPKAAMPPQIIQFPEDQKVRAGESVELFGKVTGTQPITCTWMKFRKQIQESEHMK
VENSENGSKLTILAARQEHCGCYTLLVENKLGSRQAQVNLTVVDKPDPPAGTPCASDIRSSSLTLSWYG
SSYDGGGSAVQSYSIEIWDSANKTWKELATCRSTSFNVQDLLPDHEYKFRVRAINVYGTSEPSQESE
LTTVGEKPEEPKDEVEVSDDDEKEPEVDYRTVTINTEQKVSDFYDIEERLGSGKFGQVFRLVEKKTRK
VWAGKFFKAYSAKEKENIRQEISIMNCLHHPKLVQCVDAFEEKANIVMVLEIVSGGELFERIIDEDFELTE
RECIKYMRQISEGVEYIHKQGIVHLDLKPENIMCVNKTGTRIKLIDFGLARRLENAGSLKVLFGTPEFVAPEVI
NYEPIGYATDMWSIGVICYILVSGLSPFMGDNDNETLANVTSATWDFDDEAFDEISDDAKDFISNLLKKD
MKNRLDCTQCLQHPWLMKDTKNMEAKKLSKDRMKKYMARRKWQKTGNAVRAIGRLSSMAMISGLSG
RKSSTGSPTSPLNAEKLESEEDVSQAFLEAVAEEKPHVKPYFSKTIRDLEVVEGSAARFDCKIEGYPDPE
VVWFKDDQSIRESRHFQIDYDEDGNCSLIISDVCGDDDAKYTCKAVNSLGEATCTAELIVETMEEPSSG
SGGDSLEFIASKLASGLRGSHHHHHH

## Acknowledgements

The authors acknowledge Lukas Milles and Stefan Stahl for helpful discussions. We are greatly indebted to Ms Birgit Brandmeier for initial work on protein purification, expression and activity assays. Support for this work was provided by the Deutsche Forschungsgemeinschaft SFB 863-A01 to HEG, the ERC Advanced Grant CelluFuel to HEG, the MRC grant MR/J010456/1 to MG and the BHF Chair of Molecular Cardiology to MG.

## Additional information

### Funding

| Funder | Grant reference number | Author |
|---|---|---|
| Sonderforschungsbereich | SFB863-A01 | Fabian Baumann<br>Magnus Sebastian Bauer<br>Diana Angela Pippig<br>Hermann Eduard Gaub |
| European Commission | ERC Advanced Grant CelluFuel | Fabian Baumann<br>Magnus Sebastian Bauer |

| | | Diana Angela Pippig<br>Hermann Eduard Gaub |
| --- | --- | --- |
| Medical Research Council | | Martin Rees<br>Alexander Alexandrovich<br>Mathias Gautel |
| British Heart Foundation | BHF Chair of Molecular<br>Cardiology | Martin Rees<br>Alexander Alexandrovich<br>Mathias Gautel |

The funders had no role in study design, data collection and interpretation, or the decision to submit the work for publication.

## Author contributions

FB, Conceptualization, Data curation, Software, Formal analysis, Investigation, Visualization, Writing—original draft, Writing—review and editing; MSB, Data curation, Software, Formal analysis, Investigation, Visualization, Writing—review and editing; MR, Investigation, Writing—review and editing; AA, Resources, Investigation, Writing—review and editing; MG, Conceptualization, Funding acquisition, Investigation, Project administration, Writing—review and editing; DAP, Conceptualization, Resources, Supervision, Investigation, Writing—review and editing; HEG, Conceptualization, Supervision, Funding acquisition, Investigation, Project administration, Writing—review and editing

## Author ORCIDs

Magnus Sebastian Bauer, http://orcid.org/0000-0003-1357-2852
Hermann Eduard Gaub, http://orcid.org/0000-0002-4220-6088

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
