## [Decision Letter]

Thank you for submitting your article "Mechanical Force as a Functional Regulator in Smooth Muscle Myosin Light Chain Kinase" for consideration by *eLife*. Your article has been favorably evaluated by Arup Chakraborty (Senior Editor) and three reviewers, one of whom is a member of our Board of Reviewing Editors. The following individual involved in review of your submission has agreed to reveal his identity: James R Sellers (Reviewer #3).

The reviewers have discussed the reviews with one another and the Reviewing Editor has drafted this decision to help you prepare a revised submission.

Summary:

The smooth muscle myosin kinase (smMLCK) is a catalytically active serine/threonine kinase which is triggered by a conformational change upon Ca^2+^/calmodulin (Ca^2+^/CaM) binding. smMLCK shows a structural homology with the force-activated titin kinase and thus may be expected also to be regulated by mechanical pulling stress. In this paper, the authors studied the mechanical response of smMLCK applying AFM-based single-molecule force spectroscopy (SMFS) to understand how this cytoskeletal kinase is conformationally modulated by external forces. It was particularly investigated whether the presence of different ligands including ATP, Ca^2+^/CaM or RLC peptide substrate or a combination of these affect the mechanical unfolding intermediates along the unfolding pathway of smMLCK. To reveal statistically solid insights these mechanisms have been investigated via high throughput SMFS. The characteristic pattern of transitions between different smMLCK states and their variations in the presence of different substrates and ligands are presented. The results show a very interesting interaction between kinase domain and regulatory light chain (RLC) substrate in the absence of CaM, which indicates a restored substrate-binding capability due to mechanically induced removal of the auto-inhibitory regulatory region. The data also suggest that the conformational change occurs in the presence of the substrate, suggesting that the substrate cannot bind prior to the said conformational change. In addition, the increase in unfolding force shows that the substrate does bind after the conformational change. The AFM-based single-molecule force spectroscopy experiments are conducted at outstanding quality, the experimental data is carefully analyzed and interpreted. Suitable controls have been performed and support the experimental conclusions very convincingly. The paper is written very well.

Essential revisions:

The evidence for mechanical regulation of kinase activity is indirect and thus the title may be a bit presumptuous. We realize that this is a very difficult thing to actually prove and the necessary technical advances are brought up in the Discussion. Based on this, we wonder whether the title oversells the manuscript.

Some of the speculation about possible mechanoregulation of the kinase stems from early biochemical studies that showed smooth muscle MLCK could bind myosin filaments via its C-terminus and actin filaments via the N-terminus. However, the Kd for both of these bindings were in the μm range and the recent results from the Cremo and Baker labs (Hong et al., 2015) show that the kinase diffuses along actin filaments. Based on this how would the authors imagine that the kinase could be subjected to sufficient force to pull out the autoinhibited domain rather than just dissociating from actin filaments or myosin filaments if the two systems were sliding relative to one another? Perhaps catch bonds are involved?

Reviewer #1:

In this work, they present evidence that the inhibitory domain of the kinase undergoes a tension-induced conformational change that is very similar to the change that occurs upon calmodulin binding, which is a known activator. Because the N and C termini are known to bind actin filaments and xxx, respectively, it is proposed that tension may function as an additional regulator mechanism for kinase activation. The data also suggest that the conformational change occurs in the presence of the substrate, suggesting that the substrate cannot bind prior to the said conformational change. In addition, the increase in unfolding force shows that the substrate does bind after the conformational change. Overall, I find the work interesting and of high quality, and there is a good reason to suspect that force can activate the enzyme activity.

Reviewer #2:

The smooth muscle myosin kinase (smMLCK) is a catalytically active serine/therosine kinase which is triggered by a conformational change upon Ca^2+^/calmodulin (Ca^2+^/CaM) binding. smMLCK shows a structural homology with the force-activated titin kinase and thus may be expected also to be regulated by mechanical pulling stress. In this paper, the authors studied the mechanical response of smMLCK applying AFM-based single-molecule force spectroscopy (SMFS) to understand how this cytoskeletal kinase is conformationally modulated by external forces. It was particularly investigated whether the presence of different ligands including ATP, Ca^2+^/CaM or RLC peptide substrate or a combination of these affect the mechanical unfolding intermediates along the unfolding pathway of smMLCK. To reveal statistically solid insights these mechanisms have been investigated via high throughput SMFS. The characteristic pattern of transitions between different smMLCK states and their variations in the presence of different substrates and ligands are presented. The results show a very interesting interaction between kinase domain and regulatory light chain (RLC) substrate in the absence of CaM, which indicates a restored substrate-binding capability due to mechanically induced removal of the auto-inhibitory regulatory region. The AFM-based SMFS experiments are conducted at outstanding quality, the experimental data is carefully analyzed and interpreted. Suitable controls have been performed and support the experimental conclusions very convincingly. The paper is written very well. I strongly recommend publication.

Reviewer #3:

The manuscript by Baumann et al. uses AFM to unfold smooth muscle myosin light chain kinase. From the data they infer that force may be a regulator of kinase function. The data show that various domains of the kinase unfold sequentially in a reproducible manner and the authors are able to assign unfolding events to particular domains. They show that the kinase domain unfolds first followed by the FN3 domain and then the Ig domains unfold last. The evidence for mechanical regulation of kinase activity is very indirect and thus the title may be a bit presumptuous. The main data to suggests this behavior is that the S2 state is stabilized in the presence of an RLC substrate peptide.

The study is of considerable interest and is well executed.

Regarding my comment about the data not really showing mechanical activation of the enzyme, I realize that this is a very difficult thing to actually prove and the necessary technical advances are brought up in the Discussion. Based on this, I do wonder whether the title oversells the manuscript.

Some of the speculation about possible mechanoregulation of the kinase stems from early biochemical studies that showed smooth muscle MLCK could bind myosin filaments via its C-terminus and actin filaments via the N-terminus. However, the Kd for both of these bindings were in the μm range and the recent results from the Cremo and Baker labs (Hong et al., 2015) show that the kinase diffuses along actin filaments. Based on this how would the authors imagine that the kinase could be subjected to sufficient force to pull out the autoinhibited domain rather than just dissociating from actin filaments or myosin filaments if the two systems were sliding relative to one another?

Also in the cartoon in Figure 5, the authors show the while the RLC stabilizes the lower portion of the kinase domain, the upper portion (depicted as the ATP binding site) is already unfolded. This does not seem to be a mechanism for the mechanical regulation of activity.

The legend to Figure 5 does not describe the significance of the color change of the triangular motif (CaM). Does the light gray color mean that this is apo-calmodulin?

---

## [Author Response]

Essential revisions:

The evidence for mechanical regulation of kinase activity is indirect and thus the title may be a bit presumptuous. We realize that this is a very difficult thing to actually prove and the necessary technical advances are brought up in the Discussion. Based on this, we wonder whether the title oversells the manuscript.

The shown results give strong experimental indication for mechanical force being a functional regulator in smMLCK comparable to the established Ca^2+^/CaM activation pathway. However, we agree that those evidences are drawn rather indirectly by means of AFM force spectroscopy data. Although the data is conclusive and in our opinion convincing, a direct proof of the force activation cannot be given so far. To make this clear to the reader, we would therefore like to suggest the following title:

Increasing Evidence of Mechanical Force as a Functional Regulator in Smooth Muscle Myosin Light Chain Kinase (see title).

Some of the speculation about possible mechanoregulation of the kinase stems from early biochemical studies that showed smooth muscle MLCK could bind myosin filaments via its C-terminus and actin filaments via the N-terminus. However, the Kd for both of these bindings were in the μm range and the recent results from the Cremo and Baker labs (Hong et al., 2015) show that the kinase diffuses along actin filaments. Based on this how would the authors imagine that the kinase could be subjected to sufficient force to pull out the autoinhibited domain rather than just dissociating from actin filaments or myosin filaments if the two systems were sliding relative to one another? Perhaps catch bonds are involved?

Hong et al. indeed show sliding of actin on a myosin-coated surface with MLCK having µM affinity to myosin and actin. However, the measured affinity does not necessarily implicate low binding forces as could be shown in the past by other force spectroscopy studies. Other factors such as pulling geometry or force trajectory through the receptor-ligand system could be identified as much more dominant parameters. Best example is the Strep-Tag II:Strep-Tactin handle system employed in this study itself: despite a similar Kd in the µM range, we could show that rupture forces easily suffice to not only activate but also unfold the kinase and its side domains.

A crucial factor, however, is given by the timescales of the binding partners represented in the dissociation rate. The system has to form a stable linkage lasting long enough to bridge mechanical stress and therefore allow activation of the kinase domain before it dissociates again. For a system diffusing along the filaments, however, long-term association can certainly be assumed. Additional binding domains in the long isoforms of MLCK additionally increase this cytoskeletal association as shown by Kudryashov et al. in 2004 (Myosin light chain kinase (210 kDa) is a potential cytoskeleton integrator through its unique N-terminal domain).

Our hypothesis is that relative movement of actin to myosin in the muscle sets smMLCK under tension. Even for physiological forces below values of 15 pN as shown in our AFM force spectroscopy data, activation can be expected if the molecule is kept in a stretched conformation for sufficient time (comparable to a force-clamp configuration). A catch-bond mechanism might be possible but is not essential for our interpretation.

To address this we included a paragraph in the discussion section of the manuscript.